# Dynamic Switching Teacher: How to Generalize Temporal Action Detection Models

## Abstract

Temporal Action Detection (TAD) is a crucial task in video understanding, focusing on the precise identification of the onset and termination of specific actions within video sequences. Despite advancements on certain datasets, existing methods often struggle to maintain their efficacy when applied to datasets from disparate domain. In this study, we introduce, for the first time, the application of source-free domain adaptation (SFDA) techniques to the field of TAD, aiming to enhance the generalization capability of TAD models on unlabeled target datasets without access to source data. Most popular SFDA methods predominantly follow the Mean-Teacher (MT) framework and often falter due to the significant domain shift. The generation of pseudo labels by a pre-trained teacher model on the source domain can lead to a cascade of errors when these labels guide the training of a student model, potentially causing a harmful TAD feedback loop. To address this issue, we propose a novel dynamic switching teacher strategy that integrates both dynamic and static teacher models. The dynamic teacher model updates its parameters by learning knowledge from the student model. Concurrently, the static teacher model engages in periodic weight exchange with the student model, ensuring baseline performance and maintaining the quality of pseudo labels. This approach significantly mitigates the label noise. We establish the first benchmark for SFDA in TAD tasks and conduct extensive experiments across various datasets. Our method demonstrates state-of-the-art performance, substantiating the suitability of our method for TAD.

## 1 Introduction

Temporal Action Detection (TAD) is essential for understanding long-form videos, aiming to precisely identify specific actions within untrimmed videos by determining their start and end times, along with their categories. With the rapid expansion of datasets and advancements in deep learning models, TAD has achieved remarkable performance on certain datasets (Zhang et al., 2022; Liu et al., 2024; Singh et al., 2024).

However, current TAD datasets (Caba Heilbron et al., 2015; Idrees et al., 2017; Liu et al., 2022; Carreira & Zisserman, 2017b) often lack diversity in scenarios, leading to poor generalization of traditional TAD models (as shown in Fig.1(a)) when encountering unseen scenarios with domain shifts during real-world deployment. Furthermore, unlike image data, video data is more challenging and costly to annotate with an additional temporal dimension, making it impractical to label video data for every new scenario. Consequently, there is a need for models that can adapt to new scenarios without supervision. To address the challenge of domain shift, researchers have introduced Unsupervised Domain Adaptation (UDA) (Feng et al., 2021; Luo et al., 2022; Gu et al., 2024). As shown in Fig.1(b), these methods aim to fine-tune models by leveraging labeled datasets from a source domain and unlabeled datasets from a target domain, with the goal of minimizing the domain gap between the source and target domains. Methods such as adversarial learning, which align features across domains, have been employed to enhance model performance in the target domain.

Nevertheless, most UDA algorithms assume access to the source domain data, which is often unrealistic. Video data, such as the ActivityNet1.3 (Caba Heilbron et al., 2015) (700GB), involves high storage costs and slow transmission speeds, making it unrealistic to access the source dataset once the model is deployed on different devices. Additionally, transmitting the source dataset raises

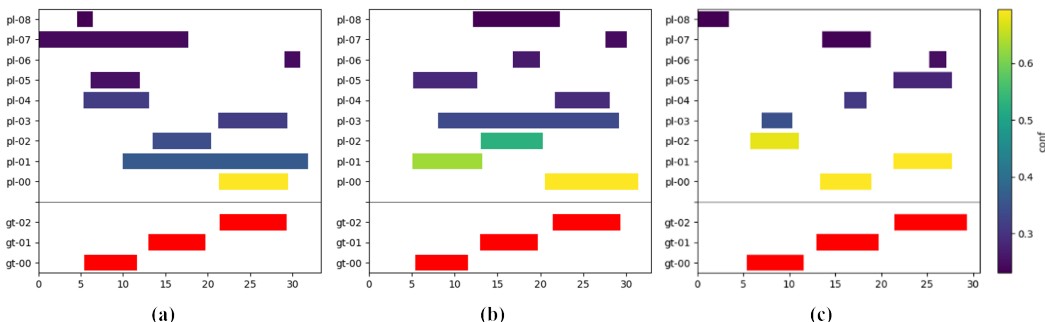

Figure 1: **Different Types of TAD Frameworks.** (a) Standard Supervised TAD Framework: the model is supervised trained using labeled images from the dataset; (b) UDA-TAD Framework: after training on the source domain, the model performs domain adaptation using both labeled source domain data and unlabeled target domain data; (c) SFDA-TAD Framework: after training on the source domain, and without access to source domain data, the model performs domain adaptation using only unlabeled target domain data.

significant privacy and security concerns. Therefore, adapting the model to a target domain without access to the source domain datasets presents a more realistic and challenging scenario. This challenge motivates our study of Source-Free Domain Adaptation (SFDA) for TAD tasks. As shown in Fig.1(c), we perform domain adaptation without accessing the source domain dataset.

Although numerous SFDA methods (Lu et al., 2023a; Yue et al., 2023a; Chu et al., 2023) have been developed for image classification and object detection, to the best of our knowledge, none have yet been applied to the task of TAD. Most existing SFDA methods (Karim et al., 2023; VS et al., 2023) follow the MT framework (Tarvainen & Valpola, 2017), utilizing a pre-trained teacher model from the source domain to generate pseudo labels for the student model. Since most previous SFDA algorithms were not designed for TAD tasks, they ignore the temporal dimension inherent in video data. As a result, directly transferring these models to TAD tasks significantly degrades performance. Additionally, the few domain adaptation algorithms tailored for videos focus on action recognition, a classification task—rather than our detection task, making them unsuitable for direct application in TAD. If these algorithms are applied directly to TAD tasks, the teacher model will generates unavoidable noisy pseudo labels, as shown in Fig.2(a). This leads the student model to learn incorrect information, thereby reducing its performance in target domain.

To address the aforementioned challenges, we propose a SFDA method for TAD based on dynamic switching teacher. Unlike the traditional MT framework, our approach introduces a multi-teacher mechanism that leverages both static and dynamic teacher models to generate pseudo labels for the student model. During training, the static teacher model iteratively updates by exchanging weights with the student model. Due to the noise of pseudo labels generated by the dynamic teacher model,

Figure 2: **Pseudo labels Generated by Different Models.** We selected certain pseudo labels to compare with the Ground Truth. The horizontal axis represents time in the video. The vertical axis pl-i represents the i-th pseudo label predict by teacher model. The color of the pseudo labels represents their confidence levels, with red indicating the Ground Truth (confidence of 1). (a) Pseudo labels generated by the source model; (b) Pseudo labels generated by the static teacher model; (c) Pseudo labels generated by the dynamic teacher model.

training the student model with these pseudo labels can lead to a decline in performance. Therefore, employing a static teacher model that retains the student model's parameters prior to training serves as an added safeguard. This new static teacher setup ensures a lower bound on the overall performance of the MT framework, thereby preventing the framework from collapsing due to the student model's failure. Additionally, since the static teacher model replicates the student model's weights, as shown in Fig.2(b), it can provide stable and reliable pseudo labels. Meanwhile, as shown in Fig.2(c), the dynamic teacher model, adapted to the target domain, generates more aggressive yet higher accuracy pseudo labels. By combining these two types of pseudo labels, we effectively reduce the noise generated by teacher models.

Since no previous work has applied SFDA to the TAD task, this paper introduces the first SFDA-TAD benchmark. We conduct experiments across three datasets, implementing and comparing some state-of-the-art SFDA methods. Our approach consistently achieves the best results across multiple experimental setups. Our contributions are summarized as follows:

- We propose the first SFDA framework for TAD, enabling the transfer of a source-domain pre-trained TAD model to a target domain without access to source data.
- We introduce a dynamic switching teacher mechanism that effectively ensures the stability of the MT framework during training, preventing potential framework collapse.
- We employ a multi-teacher fusion strategy to reduce noise in the pseudo labels generated by the teacher models.
- We establish the first benchmark for SFDA-TAD , where we implemented and compared some state-of-the-art SFDA methods. Our model consistently outperforms the others, demonstrating significant performance advantages.

## 2 RELATED WORKS

### 2.1 TEMPORAL ACTION DETECTION

TAD (Zhao et al., 2021; Pramono et al., 2022; Zhao et al., 2022a) focuses on detecting both the categories of actions and their precise start and end times within a video. It has widespread applications in areas such as abnormal behavior detection, video editing, and video summarization. Since its introduction, supervised deep learning methods for TAD have shown continuous performance improvements. Initially, actions were localized using sliding window approach (Shou et al., 2016; Yeung et al., 2016). Inspired by object detection techniques, TAD methods have been classified based on their anchor mechanisms into one-stage (Lin et al., 2017; Long et al., 2019; Sridhar et al., 2021; Yang et al., 2022b), two-stage (Chao et al., 2018; Zeng et al., 2022; Zhang et al., 2022; Zhu et al., 2023), and anchor-free approaches (Zhao et al., 2020; Lin et al., 2021; Cheng & Bertasius, 2022; Shi et al., 2023). Following the success of fully supervised TAD, researchers propose weakly supervised temporal action detection (Zhai et al., 2023; Yang et al., 2022a; Huang et al., 2022a; Ren et al., 2023; Huang et al., 2022b), which relies only on video-level category labels as supervision. More recently, semi-supervised temporal action detection has also seen rapid development, leveraging small amounts of labeled data alongside large amounts of unlabeled data, with training driven by pseudo-labeling (Xia et al., 2023; Singh et al., 2024) or consistency regularization (Ding et al., 2021; Kumar & Rawat, 2022). Although the aforementioned methods have achieved good performance on specific datasets, their models exhibit significant performance degradation when transferred to a completely new TAD dataset. Therefore, we propose using domain adaptation algorithms to enhance the generalization ability of TAD models across different datasets.

### 2.2 DOMAIN ADAPTATION

Domain adaptation focuses on adjusting models pre-trained on the source domain to reduce the domain gap, enabling them to perform effectively on the target domain. Current domain adaptation methods can be classified into two categories based on the availability of source data: UDA and SFDA.

**UDA.** Unsupervised domain adaptation focuses on reducing the domain gap when the target domain is unlabeled. The most common approaches can be categorized into two main types: adversarial

learning (Wang et al., 2021; Gao et al., 2021; Gu et al., 2024; Wang et al., 2023b) and self-training. Self-training involves training the model using pseudo labels (Feng et al., 2021; Lai et al., 2023) or consistency regularization (Luo et al., 2022; Wang et al., 2023a).

**SFDA.** Although the aforementioned UDA methods have achieved promising results, our research focuses on a more practical scenario where the source domain is unavailable (Qu et al., 2024; Luo et al., 2024; Ragab et al., 2023; Li et al., 2021; Karim et al., 2023). In such cases, Xia et al. (2021); Chu et al. (2023) addresses the challenge by dividing the target domain data into source-similar and source-dissimilar sets, allowing adversarial learning without accessing source domain data. Liang et al. (2020); Lu et al. (2023b); Yin et al. (2023) leverages the model that is trained solely on the source domain to generate pseudo labels for self-supervised training. Moreover, VS et al. (2023) introduces an instance relationship graph to guide contrastive representation learning.

**Domain Adaptation in Video.** The domain adaptation methods mentioned above are primarily designed for image-based tasks. However, directly applying image-based methods to video data without considering the spatiotemporal characteristics can lead to a significant drop in performance. To address this, Li et al. (2023) uses consistency learning from spatial, temporal, and historical perspectives to train the model. Lee et al. (2024a) proposes a global-local view alignment approach to handle temporal shifts between source and target domains in video datasets. Although these techniques have achieved advanced results for video-based domain adaptation, their focus remains on action recognition (Wu et al., 2021; Zhao et al., 2022b; Sudhakaran et al., 2023). However, temporal action detection adds the additional challenge of determining the precise start and end times of specific actions, making our work more complex and challenging compared to action recognition.

## 3 METHOD

### 3.1 PRELIMINARY

Domain adaptation tasks require a labeled source domain dataset and an unlabeled target domain dataset. We formally represent the labeled source domain data as $D_s = \{X_s^n, Y_s^n\}_{n=1}^{N_s}$, where $X_s^n$ denotes the $n^{th}$ video in the source domain. Each $X_s^n$ can be represented by a sequence of feature vectors $\{x_1, x_2, \cdots, x_T\}$ defined over discretized time steps $t = \{1, 2, \cdots, T\}$, where the total duration $T$ varies across videos. $Y_s^n$ represents the corresponding label for the input video sequence $X_s^n$, consisting of $k$ action instances $y_i$, and the number of action instances $k$ also varies across videos. Each instance $y_i = (s_i, e_i, a_i)$ is defined by its start time $s_i$, end time $e_i$ and action label $a_i$, where $s_i \in [1, T], e_i \in [1, T], a_i \in [1, \cdots, C]$ (with C being the number of action categories in the dataset). The unlabeled target domain dataset, is represented as $D_T = \{X_t^n\}_{n=1}^{N_T}$, where each $X_t^n$ corresponds to the $n^{th}$ video in the target domain without the ground-truth annotations. In contrast, the task of SFDA for TAD addresses a more practical scenario. We aim to adapt a pre-trained TAD model from the source domain to an unlabeled target domain, without utilizing any source domain data. Specifically, we aim to update the parameters of model $F$ from $\Theta_s$ to $\Theta_t$, relying solely on the unlabeled target domain dataset $D_T$, without any exposure to the source domain data.

### 3.2 DYNAMIC SWITCHING TEACHER

We denote the student, static teacher, and dynamic teacher models by $\Theta_S$, $\Theta_{ST}$ and $\Theta_{DT}$, respectively. As depicted in Fig.3, each epoch in our methodology is structured into three distinct phases: Initially, we integrate the predictions of the static teacher model and the dynamic teacher model to generate pseudo labels; subsequently, the student model leverages these pseudo labels to assimilate knowledge from the target domain and the dynamic teacher model acquires knowledge of the target domain through temporal ensembling of student model; and in the final stage, the weights of student model are exchanged with the static teacher model, ensuring that the static teacher model consistently produces stable pseudo labels, acting as a performance lower bound for the ensemble.

Consequently, our model experiences two iterative updates within a single epoch. At the beginning of the $t^{th}$ epoch, we initialize the student model, static teacher model, and dynamic teacher model as $\Theta_S^{2t}, \Theta_{ST}^{2t}$ and $\Theta_{DT}^{2t}$, respectively. They are first updated to $\Theta_S^{2t+1}, \Theta_{ST}^{2t+1}$ and $\Theta_{DT}^{2t+1}$ after applying MT using pseudo labels. Subsequently, we implement a weight exchange between the student and

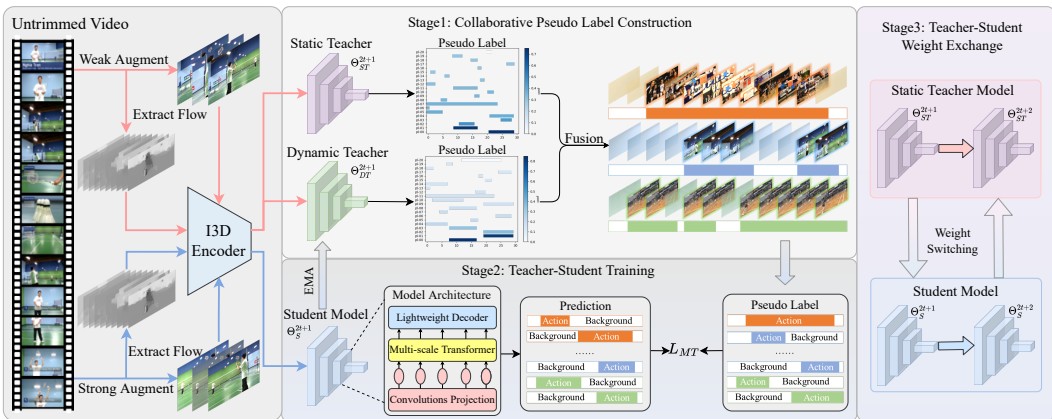

Figure 3: **Framework of Dynamic Switching Teacher.** The framework is divided into three stages, Stage One: Pseudo-label filtering and fusion; Stage Two: Training the dynamic teacher and student models; Stage Three: Weight exchange between static teacher model and student model.

static teacher model, culminating in the parameters $\Theta_S^{2t+2}$, $\Theta_{ST}^{2t+2}$ and $\Theta_{DT}^{2t+2}$, which serve as the starting point for the models in the subsequent epoch.

### 3.2.1 COLLABORATIVE PSEUDO LABEL CONSTRUCTION

In this study, we utilize a multi-teacher model framework, comprising a static teacher model, which serves as a lower bound for the entire system and generates stable pseudo labels, and a dynamic teacher model that refines its predictions through iterative learning to produce more precise and aggressive pseudo labels. Integrating predictions from both teacher models can significantly improve the quality of pseudo labels. In this section, we elaborate on the mechanisms for pseudo label filtering and the process of merging prediction from both the static and dynamic teacher models.

**Pseudo Label Filtering.** To mitigate computational demands during the teacher predictions fusion process and to prevent noise interference from low-quality predictions, we first conduct preliminary filtering of outputs from both the static and dynamic teacher models. Previous studies (Liu et al., 2023) have solely applied confidence thresholds to filter out noisy predictions. However, the static teacher model, compared to the dynamic teacher model, has less knowledge of the target domain, resulting in generally lower classification confidence. Conversely, the dynamic teacher model, having learned more about the target domain, tends to have higher classification confidence. Therefore, relying solely on confidence could lead to an overrepresentation of the dynamic teacher model's predictions in the merging process, significantly diminishing the role of the static teacher model.

To address this issue, we employ a dual-criteria approach, integrating both confidence scores and their rankings. If the number of predictions meeting the confidence threshold is fewer than $p_{min}$, we select the top $p_{min}$ predictions based on confidence ranking for filtering. If the number of predictions exceeding the confidence threshold is more than $p_{max}$, we discard those ranked beyond $p_{max}$ in terms of confidence. This approach ensures that both the dynamic and static teacher models contribute to the prediction fusion phase.

**Teacher Bounding Box Fusion.** We input weakly augmented video data into both the static and dynamic teacher models, obtaining prediction results $Y_{ST}$ and $Y_{DT}$, respectively. Both $Y_{ST}$ and $Y_{DT}$ consist of multiple predictions $(t_{ST}, c_{ST}, s_{ST})$ and $(t_{DT}, c_{DT}, s_{DT})$, where $t_{ST}$ and $t_{DT}$ denote the start and end times of the actions, $c_{ST}$ and $c_{DT}$ represent the action category, $s_{ST}$ and $s_{DT}$ indicates the classification confidence. Following the Weighted Boxes Fusion (WBF) (Solovyev et al., 2021), we select predictions with an tIoU greater than 0.5 and belonging to the same class as part of the same cluster. We compute clusters for the prediction results from both the static teacher model and the dynamic teacher model, and then fuse them to calculate the pseudo label corresponding to each cluster using the following formula:

$$\widetilde{t} = \frac{1}{S} \left( \sum_{i=1}^{M} s_{ST}^i * t_{ST}^i + \sum_{j=1}^{N} s_{DT}^j * t_{DT}^j \right), \tag{1}$$

$$\widetilde{s} = \frac{\lambda_s}{M} \sum_{i=1}^{M} s_{ST}^i + \frac{1 - \lambda_s}{N} \sum_{j=1}^{N} s_{DT}^j, \ \lambda_s \in (0, 1), \tag{2}$$

where $M$ and $N$ represent the number of predictions stored in the corresponding cluster from static and dynamic teacher models, respectively. $\lambda_s$ is a weight hyperparameter controlling the importance of the static and dynamic teacher models, and $S$ is the sum of all prediction confidences.

### 3.2.2 TEACHER-STUDENT TRAINING

Following the fusion of teacher predictions, we obtain reliable pseudo labels $\left( \widetilde{t}, \widetilde{c}, \widetilde{s} \right)$ for training the student model. By applying strong data augmentation to the target domain dataset, we obtain $D_T^{'} = \left\{ X_{aug}^n \right\}_{n=1}^{N_T}$. The loss function of student model is defined as:

$$L_{det} = \sum_{n=0}^{N_T} \lambda_d * L_{cls} \left( \Theta_S \left( X_{aug}^n \right), \widetilde{c} \right) + (1 - \lambda_d) * L_{reg} \left( \Theta_S \left( X_{aug}^n \right), \widetilde{t} \right), \ \lambda_d \in (0, 1), \tag{3}$$

where $L_{cls}$ is the focal loss function, and $L_{reg}$ is the tIoU loss function. $\lambda_d$ is a weight hyperparameter controlling the importance of $L_{cls}$ and $L_{reg}$. The student model calculates the loss based on the pseudo labels to update its parameters to $\Theta_S^{2t+1}$.

In each epoch, the static teacher model learns solely through weight exchange. To enhance the accuracy of pseudo labels in the target domain, the dynamic teacher model update its weights based on the student model. Following the conventional MT framework, we use the EMA strategy for weight updates of dynamic teacher model. Thus, in the second phase of $t^{th}$ epoch, the parameter updates for the student and dynamic teacher models can be represented as:

$$\Theta_S^{2t+1} \longleftarrow \Theta_S^{2t} + \gamma \frac{\partial \left( L_{det} \right)}{\partial \left( \Theta_S^{2t} \right)}, \tag{4}$$

$$\Theta_{DT}^{2t+1} \longleftarrow \alpha \Theta_{DT}^{2t} + (1 - \alpha) \Theta_S^{2t+1}, \tag{5}$$

where the hyperparameters $\gamma$ and $\alpha$ represent the learning rate of the student model and the EMA rate of the dynamic teacher model, respectively.

### 3.2.3 PERIODIC TEACHER-STUDENT WEIGHT EXCHANGE

During the training process, we employ a periodic weight exchange strategy to optimize the performance and stability of both the student and static teacher models. This strategy is detailed as follows: after training the student model using pseudo labels, we perform a weight exchange to allow the static teacher model to record the current weights of the student model. Meanwhile, the student model adopts the weights from the static teacher model to mitigate significant performance fluctuations during training. The exchange process can be expressed as:

$$\Theta_S^{2t+2} = \Theta_{ST}^{2t+1}, \Theta_{ST}^{2t+2} = \Theta_S^{2t+1}. \tag{6}$$

In this step, the weights of the student and static teacher models are exchanged, resulting in updated models $\Theta_S^{2t+2}$ and $\Theta_{ST}^{2t+2}$. We apply this periodic exchange throughout the entire training process, ensuring the continuity and cyclicality of training. The student model greatly benefits from this strategy. The static teacher model provides a performance guarantee, allowing the student model to recover to a more stable state through weight exchange, even if performance fluctuations occur under the guidance of the dynamic teacher model. This mechanism not only prevents rapid performance decline but also enhances the robustness of the student model.

The static teacher model plays a crucial role in this process. By periodically updating its knowledge base, the static teacher model absorbs new information at a slower pace, thereby maintaining model

stability. This slow and steady updating process helps the model maintain performance over long-term training. We construct the dynamic teacher model following the MT framework. Compared with the traditional MT, our strategy implicitly slows down the update speed of the dynamic teacher model, providing it with stronger noise resistance. Our periodic teacher-student weight exchange strategy not only prevents catastrophic forgetting and uncontrollable collapse of the student model but also ensures the stability of the teacher models and the noise resistance of the dynamic teacher model.

## 4 EXPERIMENT

### 4.1 DATASETS

In our experiments, we use three publicly available datasets: ActivityNet1.3 (Caba Heilbron et al., 2015), Thumos14 (Idrees et al., 2017), and FineAction (Liu et al., 2022). These datasets are chosen because they offer a wide range of action categories, making it easier to select similar classes for domain adaptation tasks. The detail information of these datasets is shown in Table 1.

Table 1: Summary of Datasets Used in Experiments

| Dataset | ActivityNet1.3 | FineAction | Thumos14 |
|---|---|---|---|
| Number of Videos | 13,800 | 16,732 | 413 |
| Average Video Length | 140s | 150s | 210s |
| Number of Action Categories | 200 | 106 | 20 |
| Average Actions per Video | 1.5 | 5 | 15 |
| Average Action Duration | 50s | 7s | 4s |

### 4.2 BENCHMARKS

As no SFDA benchmark specifically designed for TAD has been proposed so far, we constructed three distinct benchmark sets using the aforementioned datasets to evaluate the performance of our model.

#### 4.2.1 ACTIVITYNET1.3→THUMOS14

Both the ActivityNet1.3 and Thumos14 datasets consist of videos collected from online media, and we selected 11 shared action classes for our experiments. Despite having the same action categories, the data between the two datasets often exhibit significant domain gaps. For example, in the case of the "diving" action, the corresponding class in ActivityNet1.3 is "Springboard diving", which specifically refers to athletes diving from a springboard. In contrast, the "Diving" class in Thumos14 includes both springboard diving and high diving. This difference creates a noticeable domain gap, ideal for testing domain adaptation algorithms. Additionally, the annotation precision between ActivityNet1.3 and Thumos14 differs greatly. ActivityNet1.3 has fewer annotations per video, but each annotation covers a longer duration, whereas Thumos14 contains many dense, short annotations. This indicates that the annotations in ActivityNet1.3 are coarser, while those in Thumos14 are much more precise. The sample size per class is balanced, with about 70 videos per class in ActivityNet1.3 and 25 per class in Thumos14.

#### 4.2.2 THUMOS14→FINEACTION

Adapting from small datasets to large ones enables the use of smaller datasets to automatically annotate larger datasets, greatly reducing the cost of manual labeling. Therefore, the ability of a model to perform domain adaptation from small to large datasets is essential. In our experiments, we adapted models trained on the smaller Thumos14 dataset to the larger FineAction dataset. We collected 12 common action classes shared between the two datasets. On average, each class in the FineAction contains 147 videos, while in Thumos14, each class has only 23 videos.

Table 2: **Comparisons with Other SFDA Methods.** "Source-Only" refers to the model trained only on the source domain. We report the mAP at tIoU=0.3, 0.5 and 0.7, and the average mAP in [0.3 : 0.1 : 0.7] of each model on the three benchmarks.

| Methods | Source Free | A→T | | | | T→F | | | | F→A | | | |
|---|---|---|---|---|---|---|---|---|---|---|---|---|---|
| | | 0.3 | 0.5 | 0.7 | mAP | 0.3 | 0.5 | 0.7 | mAP | 0.3 | 0.5 | 0.7 | mAP |
| Source-Only | - | 32.8 | 18.7 | 5.5 | 18.9 | 39.2 | 23.4 | 4.2 | 22.4 | 48.0 | 32.9 | 3.7 | 27.5 |
| DANN (Ganin et al., 2016) | ✗ | 29.4 | 19.0 | 6.2 | 19.2 | 43.1 | 26.0 | 7.8 | 25.1 | 49.2 | 36.0 | 15.7 | 33.5 |
| AT (Li et al., 2022) | ✗ | 30.9 | 17.9 | 5.2 | 18.7 | 45.7 | 25.8 | 6.2 | 26.3 | 47.1 | 32.7 | 21.5 | 33.0 |
| ICON (Yue et al., 2023b) | ✗ | 29.8 | 17.4 | 4.1 | 17.2 | **48.1** | 26.9 | 8.8 | 27.2 | 49.9 | **36.3** | 20.9 | 34.2 |
| GLAD (Lee et al., 2024b) | ✗ | 32.4 | 20.3 | 6.3 | 20.3 | 46.2 | 30.1 | 10.5 | 29.0 | **52.3** | 33.8 | 18.1 | 34.9 |
| LUHP (Zhang et al., 2024) | ✗ | 30.1 | 19.7 | 6.5 | 19.6 | 46.9 | 25.5 | 7.4 | 26.2 | 49.7 | 33.2 | 18.2 | 33.7 |
| MT (Tarvainen & Valpola, 2017) | ✓ | 29.8 | 17.0 | 4.9 | 17.2 | 44.7 | 30.1 | 9.9 | 28.7 | 43.2 | 28.2 | 17.4 | 29.1 |
| SED (Li et al., 2021) | ✓ | 30.6 | 18.3 | 6.0 | 18.1 | 43.1 | 23.6 | 4.3 | 23.7 | 40.1 | 25.7 | 15.5 | 26.8 |
| $A^2$Net (Xia et al., 2021) | ✓ | 34.2 | 20.1 | 6.2 | 20.1 | 40.7 | 23.9 | 5.2 | 23.3 | 50.1 | 29.4 | 8.8 | 29.5 |
| $A^2$SFOD (Chu et al., 2023) | ✓ | **46.0** | 13.8 | 1.8 | 19.0 | 41.9 | 24.5 | 5.6 | 24.3 | 43.8 | 29.2 | 19.1 | 30.2 |
| C-SFDA (Karim et al., 2023) | ✓ | 32.1 | 18.2 | 5.5 | 18.5 | 43.4 | 27.2 | 6.3 | 26.1 | 45.9 | 31.4 | 20.1 | 32.3 |
| Ours | ✓ | 35.0 | **20.5** | **6.9** | **20.6** | 44.5 | **30.8** | **11.7** | **29.4** | 47.5 | 34.8 | **23.2** | **35.3** |

### 4.2.3 FINEACTION→ACTIVITYNET1.3

We collected 11 common action classes from the FineAction and ActivityNet1.3 datasets. Beyond the differences in action categories, these datasets also show significant variation in video characteristics and annotation granularity. For example, although the average video length is similar, 12% of FineAction videos exceed 300 seconds, while videos in ActivityNet1.3 are evenly distributed between 0 and 250 seconds. Moreover, the average number of annotations per video in FineAction is 3 to 4 times higher than in ActivityNet1.3, but each annotation in ActivityNet1.3 is 7 times longer than in FineAction. These differences in video length and annotation density create a notable domain gap between FineAction and ActivityNet1.3, making this dataset pair ideal for evaluating the model's ability to adapt across spatiotemporal domains.

## 4.3 IMPLEMENTATION DETAILS

We use Actionformer (Zhang et al., 2022) as the backbone for our TAD task. All videos are standardized to 25 FPS, and both original video frames and optical flow features are extracted. We leverage a two-stream I3D (Carreira & Zisserman, 2017a) pre-trained on Kinetics (Carreira & Zisserman, 2017b) to extract features from all videos. Since none of the datasets in our experiments are related to Kinetics, this ensures that feature extraction model do not introduce unexpected biases into the domain adaptation process. We extract 1024-dimensional features before the final fully connected layer and concatenat the I3D features from both the video and optical flow to form a 2048-dimensional input feature for the TAD model. We employ the Adam optimizer with an initial learning rate of $10^{-4}$ and apply cosine learning rate decay. The batch size is set to 16, with a weight decay of $10^{-4}$. The EMA rate in the MT is set to 0.995.

## 4.4 RESULT

### 4.4.1 BASELINE METHODS

Since no existing SFDA algorithms are specifically designed for TAD task, we implement several advanced SFDA algorithms for TAD, including $A^2$Net (Xia et al., 2021), $A^2$SFOD (Chu et al., 2023), SED (Li et al., 2021) and C-SFDA (Karim et al., 2023). In addition, we select five advanced UDA methods to compare with our method. To demonstrate the effectiveness of our dynamic teacher models, we compare them with the standard Mean-Teacher. We also conduct a baseline experiment where the model is trained solely on the source domain, representing the lower performance limit of the task.

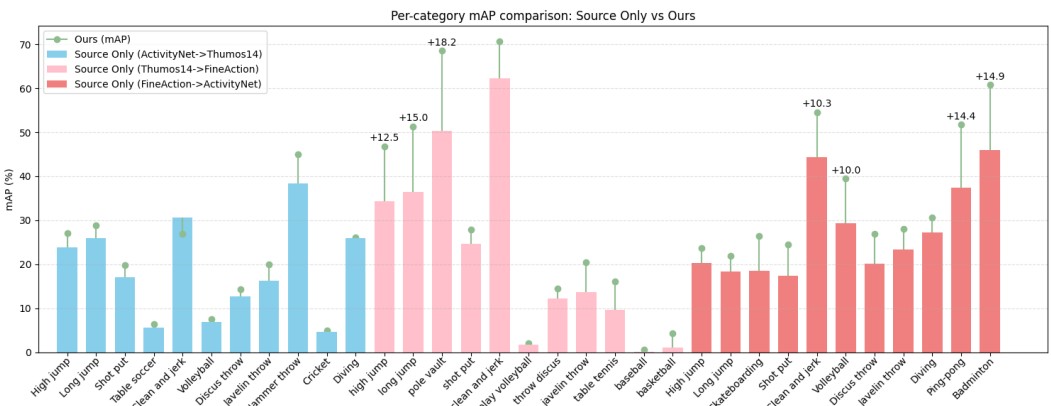

Figure 4: **Per-category mAP Comparison: Source Only vs Ours.** We report the per-category mAP for TAD under different domain adaptation scenarios. We compare the results of our method (green dots) with source-only model across three benchmarks. The blue,pink and red bars represent source-only performance in different benchmarks.

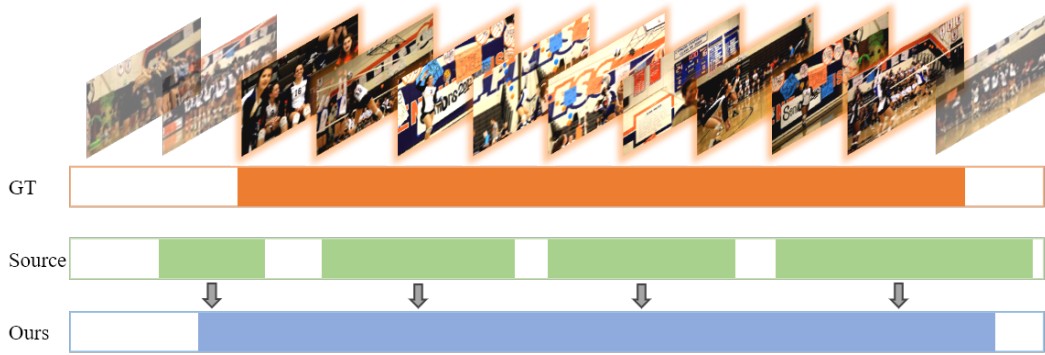

Figure 5: **Comparison of prediction density between the source-only model and our model.** The orange bar represents the ground truth time intervals where actions occur in the video. The green bar indicates the action intervals predicted by the Source-Only model, and the blue bar shows the predicted action intervals from our proposed model.

### 4.4.2 COMPARISONS WITH OTHER SFDA METHODS

We evaluate several existing SFDA methods on the three benchmarks we establish: ActivityNet1.3→Thumos14 (A→T), Thumos14→FineAction (T→F), and FineAction→ActivityNet1.3 (F→A). The results are shown in Table 2.

**ActivityNet1.3→Thumos14 (Category Difference Domain Adaptation).** We conduct experiments on 11 shared classes between the ActivityNet1.3 and Thumos14 datasets. Although these datasets contain similar categories, but they often have hierarchical relationships. For example, in the case of diving, ActivityNet1.3 includes the class "Springboard diving", whereas Thumos14 uses the broader category "Diving". Since the model is trained only on "Springboard diving" in the source domain, its ability to generalize and adapt to the broader "Diving" class in the target domain is a crucial test of its adaptability. Additionally, Thumos14 also includes certain classes that are subsets of more general actions in ActivityNet1.3. For instance, in the "Cricket" category, Thumos14 contains only the specific action "CricketShot", which challenges the model's ability to adapt to precise actions within a broader category.

We evaluate the domain adaptation performance for each action class and compare the accuracy of action detection after domain adaptation with the "Source-Only" model. As shown in Fig.4, all action classes exhibit improved performance after domain adaptation. Moreover, as indicated in Table 2, our domain adaptation method consistently outperforms other comparative approaches.

Table 3: **Ablation Study on Different Components of Our Method.** Here, PLF denotes Pseudo Label Filtering, and BBF denotes Bounding Box Fusion.

| Methods | PLF | BBF | A→T | T→F | F→A | AVG |
|---|---|---|---|---|---|---|
| Source Only | - | - | 18.9 | 22.4 | 27.5 | 22.9 |
| Mean-Teacher | - | - | 17.2 | 28.7 | 29.1 | 25.0 |
| Mean-Teacher | ✓ | - | 19.5 | 29.0 | 33.7 | 27.4 |
| Dynamic-Teacher | - | - | 18.7 | 28.5 | 32.5 | 26.6 |
| Dynamic-Teacher | ✓ | - | 18.8 | 29.1 | 34.3 | 27.4 |
| Dynamic-Teacher | - | ✓ | 18.0 | 28.9 | 33.1 | 26.7 |
| Ours | ✓ | ✓ | **20.1** | **29.4** | **35.3** | **28.3** |

**Thumos14→FineAction (Small to Large Dataset Domain Adaptation).** Transferring a model trained on a small dataset to a larger one can significantly enhance its ability to automatically annotate the larger dataset. In our experiments, we train the model on the relatively small Thumos14 dataset and then adapt it to the larger FineAction dataset. As shown in Table 2, our method achieves a mean Average Precision (mAP) of 29.37%, surpassing all comparison methods. This results in a 6.97% improvement over the source-domain pre-trained model and a 3.27% increase compared to the best-performing comparison method. Additionally, as illustrated in Fig. 4, all action categories show significant performance improvements after domain adaptation, with particularly notable gains in actions such as "High jump", "Long jump" and "Pole vault".

**FineAction→ActivityNet1.3 (Annotation Density Domain Adaptation).** The FineAction and ActivityNet1.3 datasets differ significantly in their intrinsic properties (e.g., video duration) and annotation density. When a model trained on FineAction is directly transferred to ActivityNet1.3, it tends to produce overly dense predictions, which is unsuitable for ActivityNet1.3, where annotations are sparser but cover broader time ranges. Therefore, this set of experiments tests the ability of domain adaptation algorithms to handle these differences in annotation density.

As shown in Fig. 5, the model trained only on the source domain predict multiple short segments for a single action in ActivityNet1.3. However, after domain adaptation using our algorithms, the model is able to predict a single complete action, closely matching the ground truth. Additionally, as shown in Table 2, after domain adaptation, the model achieves mAP of 35.33%, an improvement of 7.86% compared to before adaptation, and a 3% increase over the best competing method.

### 4.4.3 ABLATION STUDIES

In Table 3, we report the ablation study results for different components of our domain adaptation algorithm. First, we train the model solely on the source domain and test it on the three benchmarks, achieving an average mAP of just 25.7%. By applying the basic Mean-Teacher for domain adaptation, the average mAP increase to 27.4%. Introducing our proposed pseudo-label filtering algorithm further improve the mAP to 30.8%. Replacing the Mean-Teacher with our propose Dynamic-Teacher lead to an additional 2.5% increase in mAP. Finally, using both the pseudo label filtering algorithm and the bounding box fusion strategy, the mAP improved by 5.9% compared to the source-only model, and by 4.2% over the basic Mean-Teacher.

## 5 CONCLUSION

In this work, we propose a source-free domain adaptation algorithm for temporal action detection based on dynamic teacher switching. By employing multi-teacher collaborative training and joint pseudo label generation, our method effectively improves the stability of SFDA algorithms and enhances the generalization capability for TAD tasks. As the first work to introduce a SFDA algorithm for TAD, we also present three practically meaningful benchmarks base on current popular video datasets. We reproduce several advanced SFDA algorithms on these benchmarks, and the experimental results demonstrate that our SFDA algorithm, specifically designed for TAD tasks, outperforms previous SFDA methods.

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

# A   APPENDIX

## A.1   ADDITIONAL EXPERIMENTAL DETAILS

**Dataset Details.** The FineAction and ActivityNet1.3 datasets exhibit significant differences in annotation density, as illustrated in Fig. 6. For the same action class, the two datasets vary considerably in terms of scene context. Specifically, the FineAction dataset records brief scene transitions or irrelevant footage as background during annotation, whereas the ActivityNet1.3 dataset does not employ such meticulous labeling. Instead, it focuses solely on detecting the start and end times of actions, disregarding transitional animations or camera movements that may cause the primary subject of the action to disappear from view.

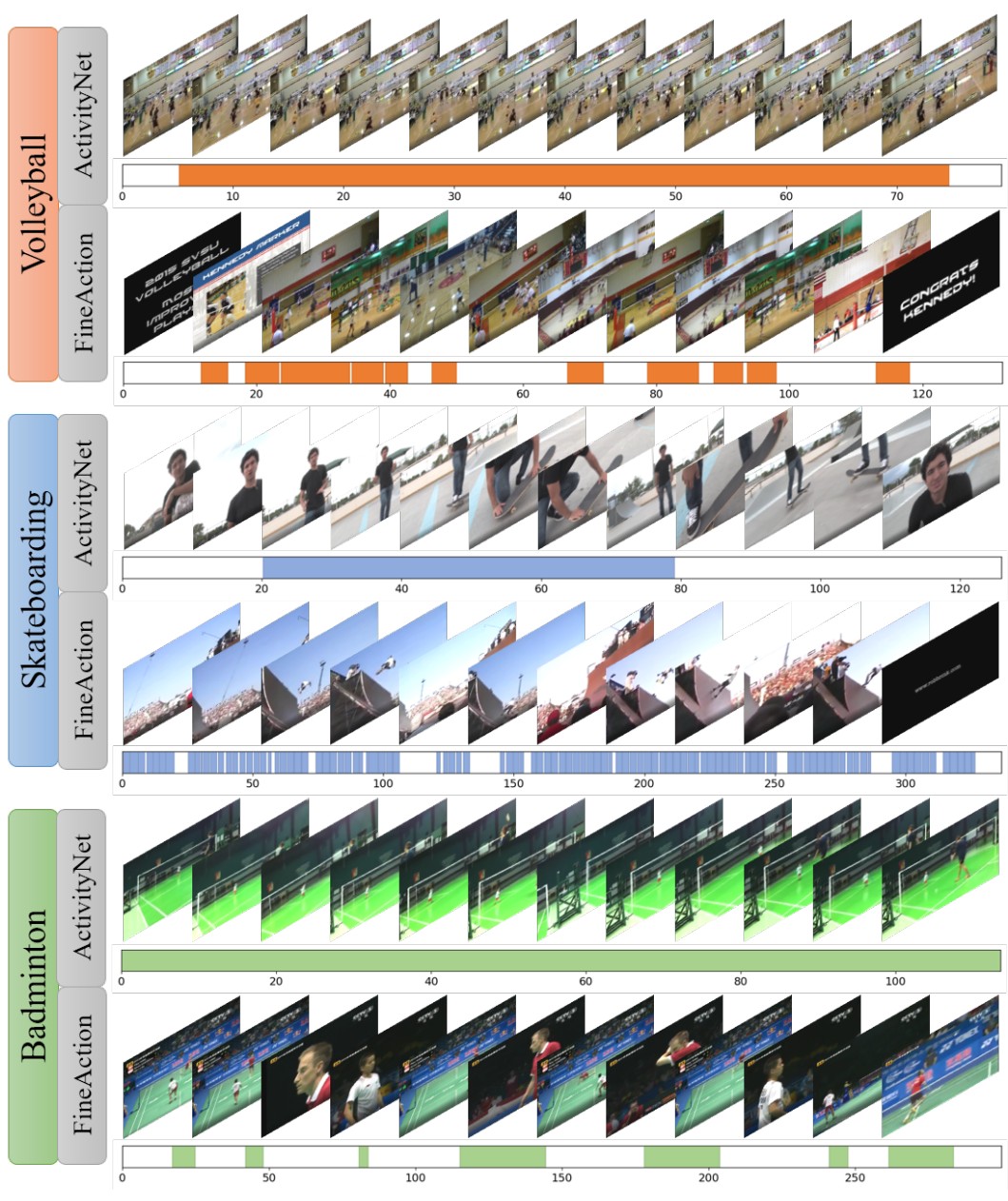

Figure 6: **Comparison of Annotation Density and Domain Gaps between FineAction and ActivityNet1.3 Datasets.**

**Evaluation Metrics.** In our experiments, we used the temporal Intersection over Union (tIoU) as the evaluation metric for temporal action detection, calculated as follows:

$$tIoU_i^j = \frac{\left(t_i^j \cap \hat{t_i^j}\right)}{t_i^j \cup \hat{t_i^j}}, \tag{7}$$

where $t_i^j$ denotes the segment corresponding to the $j^t h$ action instance in video $i$, and $\hat{t_i^j}$ represents the predicted segment for the same instance by our model.

The tIoU loss function is computed using the following formula:

$$L_{reg} = \frac{1}{n * m} \sum_{i=0}^{n} \sum_{j=0}^{m} \left(1 - tIoU_i^j\right), \tag{8}$$

where $n$ represents the total number of videos in the dataset, and $m$ indicates the number of action instances contained within each video. We employed five tIoU thresholds of 0.3, 0.4, 0.5, 0.6 and 0.7, with the final mean Average Precision (mAP) calculated as the average of the AP values corresponding to these thresholds.

**Implementation Details.** All videos from the three datasets were standardized to 25 FPS. When extracting I3D features, we set the sliding window size to 16, extracting features every 16 frames, with the stride of the sliding window also set to 16. Each frame was resized to 244x244 pixels, and optical flow features were extracted accordingly.

We conducted our experiments using five NVIDIA RTX 3080 GPUs, and the existing code used included implementations for Actionformer and I3D. The datasets utilized in our experiments were ActivityNet1.3, Thumos14, and FineAction.

## A.2 ADDITIONAL EXPERIMENTAL RESULTS

We recorded the Average Precision (AP) for all action classes in each experiment, as summarized in the table below. In addition to the results for the Thumos14-FineAction benchmark, we included an additional set of experiments for FineAction-Thumos14 to evaluate the model's domain adaptation capabilities on this benchmark.

## A.3 CODE AVAILABILITY

Code is available in Supplementary Material.

Table 4: Performance Comparison on the FineAction→ActivityNet1.3 Benchmark for Each Action Class.

| Methods | High jump | Long jump | Skateboarding | Shot put | Clean and jerk | Volleyball |
|---------|-----------|-----------|---------------|----------|----------------|------------|
| Source | 20.3 | 18.4 | 18.5 | 17.3 | 44.3 | 29.4 |
| MT | 20.5 | 18.6 | 17.7 | 19.4 | 49.7 | 33.3 |
| A$^2$Net | 19.3 | 16.7 | 20.5 | 22.6 | 48.1 | 33.0 |
| A$^2$SFOD | 21.1 | 18.4 | 21.0 | 20.1 | 50.3 | 35.2 |
| SED | 19.7 | 18.2 | 16.6 | 17.3 | 47.4 | 30.6 |
| CSFDA | 23.5 | 18.4 | 19.5 | 21.1 | 52.7 | 36.0 |
| Ours | **23.75** | **21.85** | **26.48** | **24.4** | **54.56** | **39.44** |

| Methods | Discus throw | Javelin throw | diving | Ping-pong | Playing badminton | mAP |
|---------|--------------|---------------|--------|-----------|-------------------|-----|
| Source | 20.1 | 23.3 | 27.3 | 37.4 | 45.9 | 27.4 |
| MT | 19.0 | 25.9 | 26.3 | 36.5 | 53.8 | 29.1 |
| A$^2$Net | 20.6 | 22.8 | 29.6 | 39.0 | 52.0 | 29.3 |
| A$^2$SFOD | 20.0 | 25.3 | 27.2 | 40.2 | 54.0 | 30.0 |
| SED | 16.3 | 24.6 | 23.6 | 34.4 | 45.9 | 26.8 |
| CSFDA | 21.1 | 26.5 | 28.5 | 47.6 | **63.3** | 32.2 |
| Ours | **27.0** | **28.1** | **30.6** | **51.8** | 60.8 | **35.0** |

Table 5: Performance Comparison on the FineAction→Thumos14 Benchmark for Each Action Class.

| Methods | high jump | long jump | pole vault | shot put | clean and jerk | play volleyball | throw discus |
|---|---|---|---|---|---|---|---|
| Source | 45.8 | 49.9 | 28.0 | **11.1** | **57.4** | 12.6 | 19.8 |
| MT | 62.5 | 59.4 | 43.7 | 10.8 | 39.8 | 10.8 | 31.0 |
| A$^2$Net | 49.3 | 51.6 | 44.4 | 9.3 | 51.4 | 7.3 | 31.1 |
| A$^2$SFOD | 58.9 | 60.5 | 44.7 | 4.4 | 39.8 | 16.4 | 29.5 |
| SED | 67.1 | 61.0 | **50.8** | 6.0 | 52.0 | **19.3** | 29.9 |
| C-SFDA | 53.5 | 60.2 | 43.5 | 4.6 | 50.2 | 13.4 | **33.5** |
| Ours | **73.7** | **64.0** | 34.1 | 9.3 | 55.6 | 17.7 | 30.6 |

| Methods | javelin throw | diving | table tennis | baseball | basketball | mAP |
|---|---|---|---|---|---|---|
| Source | 48.8 | 29.0 | 14.8 | 23.0 | 28.4 | 30.7 |
| MT | 61.9 | 33.3 | 15.9 | 33.6 | 27.7 | 35.9 |
| A$^2$Net | 49.9 | 26.5 | 20.1 | 32.2 | 26.3 | 33.3 |
| A$^2$SFOD | 59.1 | 33.9 | 24.9 | 24.1 | 24.9 | 35.1 |
| SED | 66.2 | 37.0 | 23.6 | 32.2 | 23.2 | 39.0 |
| C-SFDA | 59.4 | **38.0** | **26.4** | 37.0 | 23.9 | 37.0 |
| Ours | **67.1** | 23.9 | 20.7 | **42.1** | **35.2** | **39.5** |

Table 6: Performance Comparison on the Thumos14→FineAction Benchmark for Each Action Class.

| Methods | high jump | long jump | pole vault | shot put | clean and jerk | play volleyball |
|---|---|---|---|---|---|---|
| Source | 34.3 | 36.4 | 50.3 | 24.7 | 62.3 | 1.8 |
| MT | 44.5 | 49.8 | 64.9 | 28.4 | 70.6 | 1.9 |
| A$^2$Net | 34.5 | 43.2 | 46.1 | **31.8** | 68.1 | 1.9 |
| A$^2$SFOD | 34.8 | 42.3 | 49.8 | 23.9 | 61.5 | **2.2** |
| SED | 31.3 | 38.9 | 46.7 | 30.7 | 65.3 | 1.6 |
| C-SFDA | 41.0 | 48.9 | 56.2 | 28.5 | 65.3 | 2.0 |
| Ours | **46.8** | **51.3** | **68.6** | 27.8 | **70.7** | 2.0 |

| Methods | throw discus | javelin throw | table tennis | baseball | basketball | mAP |
|---|---|---|---|---|---|---|
| Source | 12.2 | 13.7 | 9.6 | 0.1 | 1.0 | 22.4 |
| MT | 14.8 | 20.0 | **16.8** | 0.5 | 3.9 | 28.7 |
| A$^2$Net | 14.7 | 14.7 | 12.0 | 0.1 | 1.0 | 24.3 |
| A$^2$SFOD | 14.2 | 16.0 | 13.4 | 0.1 | 2.4 | 23.7 |
| SED | 13.3 | 15.9 | 11.2 | 0.1 | 1.1 | 23.3 |
| C-SFDA | **15.7** | 16.6 | 11.8 | 0.1 | 0.7 | 26.1 |
| Ours | 14.4 | **20.5** | 16.0 | **0.6** | **4.3** | **29.4** |

Table 7: Performance Comparison on the ActivityNet1.3→Thumos14 Benchmark for Each Action Class.

| Methods | High jump | Long jump | Shot put | Table soccer | Clean and jerk | Volleyball |
|---|---|---|---|---|---|---|
| Source | 23.9 | 26.0 | 17.1 | 5.6 | **30.6** | 6.9 |
| MT | 21.4 | 25.0 | 15.3 | 4.7 | 29.4 | 5.9 |
| A$^2$Net | 24.9 | 28.2 | 18.0 | 6.1 | 32.4 | 7.2 |
| A$^2$SFOD | **38.5** | 31.5 | 13.9 | **8.3** | 14.9 | **15.3** |
| SED | 25.4 | **33.2** | 14.8 | 4.8 | 29.3 | 6.9 |
| C-SFDA | 23.5 | 25.4 | 16.5 | 5.5 | 29.7 | 6.8 |
| Ours | 27.0 | 28.8 | **19.7** | 6.3 | 26.8 | 7.6 |

| Methods | Discus throw | Javelin throw | Hammer throw | Cricket | diving | mAP |
|---|---|---|---|---|---|---|
| Source | 12.7 | 16.3 | 38.4 | 4.7 | 25.9 | 18.9 |
| MT | 11.1 | 14.4 | 34.4 | 4.3 | 22.8 | 17.2 |
| A$^2$Net | 13.3 | 17.8 | 39.8 | 5.0 | **27.9** | 20.1 |
| A$^2$SFOD | **16.8** | 17.0 | 11.9 | **13.4** | 27.7 | 19.0 |
| SED | 11.4 | 17.8 | 25.8 | 4.8 | 25.4 | 18.1 |
| C-SFDA | 12.8 | 16.2 | 37.4 | 4.6 | 25.5 | 18.5 |
| Ours | 14.4 | **19.9** | **45.0** | 4.9 | 26.1 | **20.6** |

