# OpenReview forum: "Dynamic Switching Teacher: How to Generalize Temporal Action Detection Models"
_ICLR.cc/2025/Conference — ICLR 2025 Conference Withdrawn Submission_

### Official Review · Reviewer_VGvu · 2024-10-31

**Soundness:** 1
**Presentation:** 2
**Contribution:** 1
**Rating:** 3
**Confidence:** 4

**Summary:**

The paper explores the adaptation of Source-Free Domain Adaptation (SFDA) techniques to Temporal Action Detection (TAD). The motivation is clear, aiming to improve model generalization across domains. The proposed method offers modest novelty, which could be acceptable if the experimental evaluations were robust enough to establish a new baseline for future research. However, the experimental setup and evaluation protocols raise significant concerns about validity.

**Strengths:**

1. The paper introduces SFDA to the TAD domain.

2. The motivation to enhance the generalization of TAD models using SFDA is well-founded and relevant.

**Weaknesses:**

1. Experimental Validity:

The paper's ultimate goal is to enhance "generalization" through SFDA. However, the experiments are conducted under a "closed-vocabulary" setting, where only videos containing the same action categories are selected across domains. This setup falls short of true generalization, especially in light of recent advances in Video-Language Models (VLMs) that push the boundaries beyond class-level generalization. The paper's evaluation protocol is thus less challenging and not sufficiently rigorous to serve as a meaningful benchmark for TAD generalization.

Here are some relevant prior works:

Chen Ju, Tengda Han, Kunhao Zheng, Ya Zhang, and Weidi
Xie. Prompting visual-language models for efficient video
understanding. In ECCV, pages 105–124. Springer, 2022.

Zhiheng Li, Yujie Zhong, Ran Song, Tianjiao Li, Lin Ma,
and Wei Zhang. Detal: Open-vocabulary temporal action
localization with decoupled networks. TPAMI, 2024.

Hyun, J., Han, S. H., Kang, H., Lee, J. Y., & Kim, S. J. (2024). Exploring Scalability of Self-Training for Open-Vocabulary Temporal Action Localization. arXiv preprint arXiv:2407.07024.



2. Conceptual Mismatch in Action Annotations:

A fundamental issue arises from the assumption that identical class labels across datasets (e.g., "cooking") denote the same action semantics. Differences in annotation granularity between datasets like Thumos and ActivityNet could mean that the same label refers to actions of vastly different temporal spans and content. For example, "cooking" might represent short clips of slicing in one dataset and an entire cooking process in another. This discrepancy poses a challenge in the SFDA setting,as the model cannot account for such variations without at least one labeled instance from the target dataset.

**Questions:**

Please see the weakness section

---

### Official Review · Reviewer_EKkF · 2024-11-02

**Soundness:** 2
**Presentation:** 2
**Contribution:** 2
**Rating:** 5
**Confidence:** 5

**Summary:**

The paper proposes a source-free domain adaptation (SFDA) method for temporal action detection (TAD), which is the first work to combine SFDA with TAD, and proposes 3 benchmarks. The proposed method is based on dynamic teacher switching, outperforming previous SFDA methods.

**Strengths:**

1. This paper is well-motivated since source-free domain adaptation is a much more practical setting compared with other domain adaptation scenarios.

2. The high-level concept of the proposed method is reasonable and easy to follow.

3. The experimental results demonstrate the effectiveness of each component of the proposed method to some extent.

**Weaknesses:**

1. Novelty/Contribution/Claim concerns:

Relationship between the proposed method and the temporal action detection problem: It is not clear how the proposed “dynamic switching teacher” method is related to this video problem. It is reasonable that this method may address the source-free domain adaptation problem, but the method is not designed for video problems. If the encoder is replaced with an image encoder, and the teacher/student models are replaced with other image-related heads, the overall pipeline could be applied to image DA problems as well. There is no component in the proposed method related to temporal information. Therefore, it is better to revise the whole motivation and claim.

More specifically, it would be better to:
* Clarify how their dynamic switching teacher approach specifically addresses challenges unique to temporal action detection, such as handling variable-length sequences or modeling temporal dependencies.
* Explain how their method differs from or improves upon applying standard image-based domain adaptation techniques to this video task.
* Discuss any components of their method that are specifically designed to handle temporal information in videos.

---

2. Technical detail concerns:

(a) pretrained teacher model from the source domain (line 078): this part is not clear. The paper only mentions that the features are extracted by a Kinetics-pretrained I3D model, but the information about TAD models is missing, including how they are initialized (line 215) and what the architecture they are (Section 4.3). It is also better to do ablation study to verify whether the proposed method is robust to the choice of pretrained source models instead of relying on a specific pretrained model.

In summary, it would be better to:
* Provide specific details on the architecture and initialization of the TAD models used.
* Clarify how the Kinetics-pretrained I3D model is integrated into their overall framework.
* Conduct an ablation study using different pretrained source models to demonstrate the robustness of their method.

(b) Sec. 3.2 dynamic switching teacher: it is not clear why MT would update static teacher (line 215), and it is also not clear why weight exchange would update dynamic teacher (line 236). In addition, the above statement is not aligned with Figure 3.

(c) Sec. 3.2.2 Teacher-Student Training: the notation is confusing. It seems that the pseudo label ($\tilde{t}$, $\tilde{c}$, $\tilde{s}$) is for one video, but $\tilde{t}$ and $\tilde{s}$ are for one cluster only in Eq. (1) and Eq. (2). Moreover, the sentence “In each epoch, the static teacher model learns solely through weight exchange” (line 295) is weird because Sec. 3.2.2 should be only for dynamic teacher.

---

3. Experiment concerns:

(a) Table 2: Since this paper claims that it is the first work about source-free domain adaptation for temporal action detection, it needs to be compared with some previous source-free video domain adaptation methods to justify that previous methods cannot be directly applied to the new problem defined in this paper. At least STHC (mentioned in Section 2 but not here), ATCoN [12], and some domain adaptive action recognition methods listed below (see “Missing citations”) are missing.

To improve this part, it would be better to:
* Include comparisons with STHC, ATCoN, and other relevant domain adaptive action recognition methods in Table 2.
* Provide a brief discussion on why these methods may not be directly applicable to temporal action detection, highlighting the unique challenges of TAD that their method addresses.
* If possible, implement or adapt some of these methods for TAD to provide a more comprehensive comparison.

(b) Implementation Details: I3D is too old. Unless there are specific reasons, a transformer-based method (e.g., MViT [13], VideoSwin [14]) should be adopted as the backbone.

To improve this part, it would be better to:
* Justify the choice of I3D, if there are specific reasons for using it.
* Conduct additional experiments using MViT or VideoSwin as the backbone to demonstrate the generalizability of their method across different architectures.
* Discuss how the choice of backbone might impact the performance of their method in the context of SFDA for TAD.

(c) Section 4.4.3: the numbers in the context (lines 522-529) are not aligned with Table 3. For example, the numbers 25.7 and 30.8 are not in Table 3.

---

4. Writing concerns:

(a) line 086-116: it is not clear what the dynamic teacher is doing in this paragraph if not reading Section 3.2. However, it is better to briefly introduce it in Section 1 since it is the crucial component of the proposed method.

(b) Figure 2 is not clear:
* Why are there multiple ground truth labels in one video?
* What are the relations between different pl-i? Are they ordered by confidence?

---

5. Missing citations:

(a) Temporal Action Detection: action segmentation is highly related to temporal action detection, and sometimes the approaches could be interchangeable. Therefore, it is better to mention action segmentation in Section 2 and at least cite some classic papers, such as MS-TCN [1], SSTDA [2], FINCH [3], HASR [4], ABS [5], and LTContext [6].

(b) Domain Adaptation in Video: there are a lot of missing citations here. Since this paper is about video with domain adaptation, some classic papers (not only GLAD) need to be cited, such as TA3N [7], MM-SADA [8], SAVA [9], CoMix [10], and TranSVAE [11]. Moreover, this paper is highly related to source-free video domain adaptation, so all the related papers should be cited, but ATCoN [12] is missing.

References:
* [1] Yazan Abu Farha, et al. "MS-TCN: Multi-stage temporal convolutional network for action segmentation", CVPR, 2019.
* [2] Min-Hung Chen, et al. "Action segmentation with joint self-supervised temporal domain adaptation", CVPR, 2020.
* [3] M. Saquib Sarfraz, et al. "Temporally-Weighted Hierarchical Clustering for Unsupervised Action Segmentation", CVPR, 2021.
* [4] Hyemin Ahn, et al. "Refining Action Segmentation with Hierarchical Video Representations", ICCV 2021.
* [5] Guodong Ding, et al. "Leveraging Action Affinity and Continuity for Semi-supervised Temporal Action Segmentation", ECCV, 2022.
* [6] Emad Bahrami, et al. "How Much Temporal Long-Term Context is Needed for Action Segmentation?", ICCV, 2023.
* [7] Min-Hung Chen, et al. "Temporal Attentive Alignment for Large-Scale Video Domain Adaptation", ICCV, 2019.
* [8] Jonathan Munro, et al. "Multi-Modal Domain Adaptation for Fine-Grained Action Recognition", CVPR, 2020.
* [9] Jinwoo Choi, et al. "Shuffle and Attend: Video Domain Adaptation", ECCV, 2020.
* [10] Aadarsh Sahoo, et al. "Contrast and Mix: Temporal Contrastive Video Domain Adaptation with Background Mixing", NeurIPS, 2021.
* [11] Pengfei Wei, et al. "Unsupervised Video Domain Adaptation for Action Recognition: A Disentanglement Perspective", NeurIPS, 2023.
* [12] Yuecong Xu, et al. "Source-free Video Domain Adaptation by Learning Temporal Consistency for Action Recognition", ECCV, 2022.
* [13] Haoqi Fan, et al. “Multiscale Vision Transformers”, ICCV, 2021.
* [14] Ze Liu, et al. “Video Swin Transformer”, CVPR, 2022.

**Questions:**

The main issue of this paper is that the proposed method is not fully justified, especially in the following two aspects:

1. The connection between the proposed method and the problem (SFDA + TAD) is not strong. It is not clear whether the proposed dynamic switching teacher is important in addressing the temporal action detection problem.

2. There are too many missing comparisons, especially for video domain adaptation methods. Since action recognition is simpler than the TAD problem, it would be better to show that directly applying previous video domain adaptation techniques is not enough the address the new problem (i.e., SFDA + TAD) defined in this paper.

However, I still appreciate the proposed method and think it is a good direction. Therefore, I initially rated this paper as “marginally below the acceptance threshold”. To increase my rating, I expect to see my main concerns being addressed in the rebuttal.

---

### Official Review · Reviewer_Jesw · 2024-11-02

**Soundness:** 2
**Presentation:** 2
**Contribution:** 2
**Rating:** 3
**Confidence:** 4

**Summary:**

This paper proposes a Source-Free Domain Adaptation (SFDA) approach for Temporal Action Detection (TAD), allowing TAD models to generalize across domains without needing labeled source data. Traditional methods often rely on Mean-Teacher (MT) frameworks, which struggle with large domain shifts due to error-prone pseudo labels. The authors introduce a dynamic switching teacher strategy combining both dynamic and static teacher models: the dynamic teacher learns from the student model, while the static teacher maintains baseline performance, reducing label noise. Experiments show that this dual-teacher approach achieves state-of-the-art results on various datasets, enhancing TAD model robustness in diverse scenarios.

**Strengths:**

- Clear motivation for problem: The problem statement is well-defined and effectively highlights the limitations of current TAD methods in handling domain shifts.

- Novelty: First work to introduce Source-Free Domain Adaptation (SFDA) for Temporal Action Detection (TAD).

- Clarity: The paper is generally easy to understand, with Figure 1 providing a helpful overview of the methodology.

**Weaknesses:**

- Unclear motivation for approach: The connection between the proposed dynamic switching teacher strategy and the specific challenges of TAD is unclear. The method appears quite general and not explicitly designed for TAD’s unique requirements.

- Limited dataset variety: Experiments are conducted on small datasets with few action classes, which limits the robustness and generalizability of the findings.

- Performance variation at lower thresholds: The proposed method shows noticeable performance discrepancies at lower thresholds, raising concerns about consistency across varying conditions.

- Insufficient clarity in ablations and stronger baselines: The ablation studies lack important experiments, making it difficult to assess the impact of each component in the approach.

- Novelty in approach: Dynamic weight switching is not novel, it was proposed in Liu et. al. ICCV 2023. The filtering and fusion is also almost similar, the only difference is the use of ranking, for which there is no analysis/ablation. Considering this, there is no technical novelty in this work.

**Questions:**

Clarifications:

- It is not clear how the setup was decided. There is A->T, T->F, and F->A, why T->A, F->T, and A->F were not done? There was no explanation behind this choice.

- Why the method does not work well in case of lower threshold? 0.3 and also 0.5.

- If the static model is exchanging weights from student model, why it is static? And how it is different from dynamic? At some point they will be same, since dynamic is just an ema of student. It is not clear why this approach should mitigate label noise, as claimed in the abstract. If the teacher is static, why we have to exchange the weights? The motivation is not clear.

- Table 3, it seems mean teacher with simple pseudo-label filtering performs really well, better than dynamic teacher with label filtering. It raises a concern whether dynamic teacher is useful or not.

- The summary in Table 1 is good, but the actual details should be provided which corresponds to the setup used in this work. It appears, the used datasets are very small in size (700-800 in ActivityNet, 200-330 videos in thumos, and 1500 in fineaction) and with only 11-12 action categories.

- L415: "Since none of the datasets in our experiments are related to Kinetics, this ensures that feature extraction model do not introduce unexpected biases into the domain adaptation process", this might not be true, Kinetics will have some action categories overlapping with these datasets, for example, diving, etc. And, it is also from youtube videos.

- Line 267: “we select predictions with an tIoU greater than 0.5 and belonging to the same class as part of the same cluster.” IoU of what? Both teach predictions?

- [Line 052:] ActivityNet1.3 (Caba Heilbron et al., 2015) (700GB). As per the methodology section, videos are represented as I3D features per frame. Does feature representation of videos take 700GB, that doesn’t seem right.

Missing details:

- There are no details provided for weak and strong augmentations.

- It is not clear if all the loss functions were same as baseline Mean-Teacher, for example, use of focal loss in the proposed method?

- The GFLOP/compute cost and parameter comparison is missing in Table 3. The proposed method is very expensive and hence estimating its true worth without knowing gain in computation is difficult.

Ablations and analysis:

- A comparison with traditional dual teacher setup will be a good baseline to compare with.

- Figure 4: comparison with source only is not very meaningful, adaption on target domain is expected to improve the performance. Similar plot with the baseline might be more interesting to discuss.

- Figure 5 is also not very interesting, If source data has short actions, such fragmented predictions are expected and since target data has long actions adaptation should make it long. A more interesting analysis will be to see if other baseline models are failing to do so, or what their prediction looks like and how it will be different for the proposed method.

- Pseudo label filtering is based on heuristics to filter out noisy pseudo labels, there is no analysis on the impact of those hyerparameters, how were they determined? Similarly, fusion of detections from two teachers is not analyzed in detail.

- Design choices missing: 1) Variations on “weight exchange”, 2) Variation of lambda weight of static and dynamic teacher.

Other minor points:

- A suggestion; the term teacher bounding box fusion may not be appropriate, bounding box is usually used for a spatial detection, using it for temporal boundaries is confusing and not intuitive.

---

### Official Review · Reviewer_9uEV · 2024-11-04

**Soundness:** 2
**Presentation:** 3
**Contribution:** 2
**Rating:** 5
**Confidence:** 5

**Summary:**

The paper addresses the problem of source-free domain adaptation (SFDA) for temporal action detection (TAD) for the first time. The proposed method claims to mitigate the cascading error problem of the mean teacher baseline by dynamically switching teacher model weights. Initially, it combines the predictions of both static and dynamic teachers to generate pseudolabels, which are then used to train the student model. The dynamic model functions as the mean teacher, incorporating the moving average weights of the student, while the weights of the static teacher are periodically updated with those of the student model.

**Strengths:**

- The first method to introduce source-free domain-adaptive temporal action detection.
- The paper is well-written, with strong motivation provided for the problem.

**Weaknesses:**

- **W1**: A primary issue with the work is its limited novelty and technical contribution. The method claims to employ periodic weight switching to reduce noisy labels and error propagation. However, there are multiple issues with this:

  - *W1.1*: The paper does not present solid evidence to show that weight switching effectively reduces noise in pseudolabels (i.e., improves their quality). It repeatedly asserts this without providing ablations, which weakens the credibility of the claims.
  - *W1.2*: The method lacks specific components tailored to temporal action detection. Apart from using weighted box fusion for pseudolabel fusion (borrowed from Solovyev et al., 2021), no unique contribution is made toward TAD. The proposed weight switching lacks a clear connection to the TAD task, making the training scheme unjustified. In its current form, the method lacks any training components related to the temporal dimension and could be applied to any source-free domain adaptation (SFDA) problem, such as image classification or object detection. To establish the effectiveness of dynamic weight switching for SFDA, the method should also be applied to classic image-based SFDA problems with established baselines, rather than relying solely on self-reported baselines for the new TAD task.

- **W2**: While the paper makes an effort to provide the first benchmark for the SFDA-TAD problem, the proposed benchmarks are not particularly meaningful.

  - *W2.1*: The benchmarks contain very few action classes (10-11 classes), which oversimplifies the problem and raises concerns about the benchmark's reliability. The primary objective of domain adaptation is to establish more practically generalizable and robust protocols; however, the simplicity of such limited action classes makes it difficult to draw meaningful conclusions.
  - *W2.2*: The paper lacks information on paired common action class names for each dataset transfer. Based on Fig. 4, I attempted to infer some pairs, but aside from “Diving” vs. “springboard diving,” there appears to be no other pair showing a significant shift in the ACTIVITYNET1.3 → THUMOS14 protocol. (While I understand annotation differences, they do not affect the class distribution.)

**Questions:**

Q1: See Weakness W1. Could the authors clarify what makes the method specific to TAD and not other SFDA problems? On a close reading of the paper, it seems that, aside from the borrowed weighted box fusion, there are no components unique to TAD. Please justify and provide results on image-based SFDA tasks as well. Since prior work results are self-reported, the observations remain inconclusive.

Q2: Please provide the class mapping for each dataset protocol. Given that the proposed benchmarks only include 10-11 classes, this should not be too challenging to provide.

---

### Note · Authors · 2024-11-13

I have read and agree with the venue's withdrawal policy on behalf of myself and my co-authors.